# Identification of Coronary Artery Diseases Using Photoplethysmography Signals and Practical Feature Selection Process

**DOI:** 10.3390/bioengineering10020249

**Published:** 2023-02-13

**Authors:** Amjed S. Al Fahoum, Ansam Omar Abu Al-Haija, Hussam A. Alshraideh

**Affiliations:** 1Biomedical Systems and Informatics Engineering Department, Yarmouk University, Irbid 21163, Jordan; 2Industrial Engineering Department, JUST, Irbid 22110, Jordan

**Keywords:** PPG signals, detection applications, cardiorespiratory function, feature combination, biomedical analysis, biomedical signal processing

## Abstract

A low-cost, fast, dependable, repeatable, non-invasive, portable, and simple-to-use vascular screening tool for coronary artery diseases (CADs) is preferred. Photoplethysmography (PPG), a low-cost optical pulse wave technology, is one method with this potential. PPG signals come from changes in the amount of blood in the microvascular bed of tissue. Therefore, these signals can be used to figure out anomalies within the cardiovascular system. This work shows how to use PPG signals and feature selection-based classifiers to identify cardiorespiratory disorders based on the extraction of time-domain features. Data were collected from 360 healthy and cardiovascular disease patients. For analysis and identification, five types of cardiovascular disorders were considered. The categories of cardiovascular diseases were identified using a two-stage classification process. The first stage was utilized to differentiate between healthy and unhealthy subjects. Subjects who were found to be abnormal were then entered into the second stage classifier, which was used to determine the type of the disease. Seven different classifiers were employed to classify the dataset. Based on the subset of features found by the classifier, the Naïve Bayes classifier obtained the best test accuracy, with 94.44% for the first stage and 89.37% for the second stage. The results of this study show how vital the PPG signal is. Many time-domain parts of the PPG signal can be easily extracted and analyzed to find out if there are problems with the heart. The results were accurate and precise enough that they did not need to be looked at or analyzed further. The PPG classifier built on a simple microcontroller will work better than more expensive ones and will not make the patient nervous.

## 1. Introduction

Cardiovascular diseases (CVDs) are illnesses of the heart and blood vessels that result in inadequate arterial blood flow to many essential organs [1]. Recent data worldwide shows that the number of people with heart disease is increasing. CVDs were responsible for about 30% of deaths worldwide in 2018 and about 36% of deaths in Jordan in 2020, according to the Jordanian Ministry of Health (MOH) [2]. So, sensitive, practical, and cost-effective ways to monitor CVDs early on are needed, in order to speed up efforts to stop the rise of CVD-related deaths [3]. Classification algorithms based on machine learning are commonly employed to anticipate and categorize the many features of biomedical signals and molecules, such as cardiac anomalies, toxicity, or biological activity [4,5,6,7,8]. Three datasets were generated to test different classification models based on machine learning [4]. Further, they applied many classification performance indicators from many fields. To compare the two groups, they used a novel method that was based on traditional chemometric techniques, such as the sum of ranking differences (SRD) and analysis of variance (ANOVA) [8].

Risk factors, such as smoking, obesity, high blood pressure, family history, and others, can be used to predict all cardiovascular diseases [9]. On the other hand, the goal of the healthcare revolution was to make monitoring technologies, such as PPG, that were small, cheap, and easy to use and could predict the start of CVDs. PPG is a non-invasive method of examining a specific body signal that provides detailed information about the heart and lungs. Several researchers focused on finding links between the PPG signal and events in the heart and blood vessels, so they could predict CVDs [9,10,11,12]. They did this by relying on extracting the PPG signal’s temporal domain properties [9,10,11,12]. Aboy et al. presented a pressure-based technique for automatic beat detection [13]. Solosenko et al. automatically exploited PPG signals to detect pre-ventricular contraction (PVC) [14]. Yousefi et al. [15] found a way to find PVC automatically by combining the chaotic nature of the PPG signal with higher-order statistics (HOS). They used several chaotic and statistical features, such as the Lyapunov exponent, skewness, kurtosis, fuzzy entropy, and spectral entropy, which were taken from the signals. Principal component analysis (PCA) was used to figure out which data points should be used to group the data. Poloania et al. and Abu Elhijja et al. developed algorithms for classifying cardiac arrhythmias based on PPG [16,17]. By taking PPG and ECG measurements, Nano et al. figured out the time between heartbeats and pulses [18]. Quality assessment for signal reliability estimation was proposed in [19] for PPG beat recognition and morphology estimation. Recently, Prabhakar et al. [20] developed a few metaheuristic techniques for reducing dimensionality. Post-classifiers then used the reduced values to separate PPG normal and CVD signals.

Recent strategies have focused on figuring out the best set of PPG characteristics that can be used to classify things helpfully. Data mining techniques, which are used to obtain useful information from data, can be used to find and emphasize differences in PPG time-domain characteristics between healthy human subjects and CVD patients [21]. Al-Fahoum et al. used data mining and a signal processing strategy called “multiple signal classification” to classify the difference between CVD patients and healthy normal subjects [12]. The primary applications of data mining techniques are the classification, prediction, and grouping of observations [22,23]. Improving the performance and usefulness of data mining algorithms means finding the best way to pick the most important attributes [15,24,25,26,27,28]. Recent research publications [28,29,30,31,32,33] demonstrated the need for automatic detection strategies for identifying arrhythmias using cutting-edge PPG-based techniques. The results from [28] to [33] supported the study’s goal. Based on an analysis of the time-domain features of the PPG signal, this study aims to find the best set of studied features that use data mining techniques to combine the most critical features of the PPG signal with other demographic information to tell the difference between healthy human subjects and CVD patients. 

## 2. Materials and Methods

### 2.1. Data Collection 

The primary goal of this study is to distinguish between normal and abnormal cardiovascular cases by combining time-domain features of the PPG signal with a set of subject demographic characteristics. Data from 360 people (200 healthy and 160 with CVD) were collected for solid results. As for the PPG, data collection from CVD patients was carried out at the internal medicine clinic of Princess Basma Hospital. The study focused on five common types of CVDs: acute coronary syndrome (ACS) (63 patients), cerebrovascular accident (CVA) (50 patients), deep vein thrombosis (DVT) (23 patients), heart failure (HF), and atrial fibrillation (AF) (13 patients each). A chart of the five CVD cases that were studied, each with its percentage from the CVD sample, is displayed in Figure 1. 

This study employed the PO-80 pulse oximeter from Beurer Healthcare in Germany to take non-invasive PPGs (pulse rate), SpO_2_, and heart rate measurements. PO-80 is a portable, rechargeable, small (L 57 mm × W 32 mm × H 30 mm), lightweight (about 42 g), and compact device that comes with “SpO_2_-Viewer/Manager” software that works with Windows. Using a USB connection, the software can send measured data from the device to the PC, which can be viewed in real-time. The sensor has an accuracy of +/−2% for SpO_2_ in the range of 30–100% measurement and +/−2 beats/minute for pulse rate monitoring in the range of 30–250 beats/minute, according to its specifications. 

The SpO_2_ sensor comprises an emitter and a detector housed in a finger clip probe. The emitter comprises a group of light-emitting diodes (LEDs) that give off red light at 660 nm and infrared light at 905 nm. The detector, in contrast, is a silicon-based photodiode. The method used to measure PPG is conducted by inserting one finger into the finger opening of the pulse oximeter. By pressing the function button on the PO-80 sensor, the oxygen saturation of the blood’s hemoglobin and the heart rate can be measured non-invasively [34].

### 2.2. Data Description

In addition to the PPG signal recorded for at least 2 min while the volunteer was at rest, the following ten variables about the volunteer’s background and health were also collected. 

#### 2.2.1. Demographic and Health Status Variables

##### Age

Age is considered a primary factor highly correlated with arterial stiffness, and its effect is reflected in the shape of the PPG signal [23,24].

##### Height, Weight, and Body Mass Index (BMI)

Even though BMI is an indicator of obesity that can increase the risk of many health problems, such as hypertension and diabetes, BMI can forecast the beginning of cardiovascular diseases [25,35].

##### Gender 

Gender influences several vital signs in the human body [1], and researchers in [36] discovered no binding effect for changes in PPG signals due to gender. Regitz-Zagrosek et al. [37] said that cardiovascular events and heart failure are different for men and women, and Al-Fahoum et al. [12] found that men and women had different results during and before exercise.

##### Blood Pressure

Arza et al. [38] and Samria et al. [39] found a relationship between the PPG signal and blood pressure. Blood pressure correlates well with some PPG features.

##### Respiratory Problems

Respiration is a vital sign closely linked to the heart and lungs because of interactions between the two [11]. Monitoring respiratory activity may reveal important information that may aid in diagnosing CVD [3].

##### Smoking

Clair et al. statistically analyzed the relationship between smoking and CVDs risk. Their study revealed that quitting smoking can potentially decrease CVDs risk [40].

##### Physical Activity 

Increasing physical activity improves cardiorespiratory fitness, which reflects favorably on the cardiovascular system by decreasing the risk of CVDs [41].

##### Other Variables (Diabetes, Kidney Failure, and Pregnancy)

Wannamethee et al. studied the effect of diabetes on the cardiovascular system and found that type I diabetes can influence the risk of coronary heart disease [42]. On the other hand, kidney failure and pregnancy were included in this study.

#### 2.2.2. Pulse Oximetry

##### Arterial Oxygen Saturation (SpO_2_)

SpO_2_ is between 97% and 99% in healthy individuals and is clinically acceptable at 95%. If the SpO_2_ value exceeds 95%, it may indicate an inadequate oxygen supply or hypoxia. 

##### Pulse Rate

The average pulse rate is between 60 and 100 beats per minute. Caffeine, exercise, stress, certain medications, and other things can speed or slow the pulse rate.

#### 2.2.3. PPG Signal

The PPG signal is susceptible to the subject’s movements and breathing activity. These artifacts can change the baseline of the signal, causing shifts and offsets. In order to remove different artifacts and obtain a pure pulse wave, an elliptic band pass filter with a sampling frequency of 120 Hz, a lower cut-off frequency of 0.6 Hz, and an upper cut-off frequency of 15 Hz was used to eliminate the breathing effect. 

The time-domain analysis of each pulse wave is the most important part of extracting features from the PPG signal. For this analysis, it is a must to find several time intervals and figure out where the peaks are in the signal [43]. The MATLAB software analyzed the pure PPG signal by taking out the following time-domain features:

##### Systolic Amplitude 

Figure 2 shows the systolic amplitude that is the highest and the first peak of the pulse. Elgendi [44] mentioned that the systolic amplitude was found to be directly proportional to local vascular distensibility, while Awad et al. [10] stated that finger PPG systolic amplitude has a low correlation to the systemic vascular resistance (SVR).

##### Peak-to-Peak Interval (∆T)

The time interval between two sequential systolic peaks can be seen in Figure 2. This interval is approximately the same as the R–R interval in ECG. 

##### Pulse Interval 

Elgendi defined the pulse interval as the time allocated between two consecutive minimums of the pulse [44]; see Figure 2. It relates to the time required for systole and diastole to be completed.

##### Crest Time (CT)

Crest time is the interval between the start of the pulse wave and the time at the systolic peak. Angius et al. defined it as the amount of time needed for the fast ejection phase [9].

##### Pulse Width (PW)

As seen in Figure 3, it is the entire width of the pulse at the half height of the systolic peak. Awad et al. [10] calculated the SVR depending on the pulse width of the finger and PPG signals; they suggested that the higher the pulse width, the higher the SVR.

##### Dicrotic Notch 

Figure 3 points to the dicrotic notch point separating the systolic and diastolic phases. Angius et al. stated that sometimes the second or diastolic peak is absent in the pulse wave, especially in older adults and CVDs patients; in this case, the dicrotic notch substitutes for this peak [9].

##### Diastolic Time

It is the time interval from the dicrotic notch to the end of the pulse wave; it relates to the time required for the diastolic phase to be completed. 

##### Pulse Transit Time (PTT)

Figure 3 shows the transit time of the pulse and the time between the systolic peak and the diastolic notch. According to Peulić et al. [11], age is negatively proportional to PTT. 

##### Total Area and Inflection Point Area Ratio (IPA)

The total area is the whole area under the pulse wave curve that can be divided into two areas: the systolic and diastolic areas. These two areas are separated at the dicrotic notch. The total area parameter was suggested to indicate arterial stiffness [45]. The ratio of the diastolic area-to-systolic area is called IPA, and it can be used to measure total peripheral resistance [46].

##### Augmentation Indices (AI) and Time Ratios

Table 1 shows four different augmentation indices and three different time ratios that can be calculated from the PPG waveform. Augmentation indices may be used as a measure of systemic arterial stiffness [47,48]. On the other hand, Angius et al. [9] found a higher mean value of relative crest time for CVD patients.

##### APG Parameters and Their Ratios

In Figure 4, the parameters taken from the APG signal are shown. These parameters can be used to measure vascular aging. Baek et al. [49] studied some ratios between APG parameters and found them to be related to the cardiovascular system and age.

### 2.3. Feature Selection and Classification

Selecting features or attributes is searching for and selecting the best set of features that provides the highest classification accuracy. WEKA software allows for the selection of features from the dataset by applying two-step procedures: an attribute evaluator and a search method. In this study, the classifier subset evaluator method was used as the attribute evaluator. This method uses a classifier to estimate the best set of attributes. The classifier subset evaluator figures out how valuable a subset of attributes is by looking at how well each feature predicts on its own and how much it overlaps with other features. Subsets of attributes that are highly correlated with the class, while having low inter-correlation, are preferred. At the same time, the searching method was greedy stepwise, which performed a forward or backward search through the space of attribute subsets. For each classifier, all the features were analyzed, and the set of features that maximized the accuracy of the classifier was selected. 

The frequencies of three CVD cases in the collected data, namely AF, DVT, and HF, were relatively small, compared to the other two cases, leading to an imbalanced class frequency problem. Under such situations, classifiers ignore the lower frequency classes and treat them as errors. In order to force the classifier to pay more attention to lower-frequency classes, we replicated the low-frequency classes. 

### 2.4. Machine Learning Classification Algorithm

The proposed algorithm was conducted in four steps, as shown in Figure 5. In the first step, data for healthy and CVD patients were collected. Information about the person being studied’s age, gender, health, and PPG signal were among the things that were collected. MATLAB extracted time-domain features from each PPG signal. In the third step, a feature selection-based classifier was used. This was performed using the Waikato Environment for Knowledge Analysis (WEKA) software. This step involved two stages: the first distinguishes between healthy and CVD subjects and the second classifies five CVD cases. A feature selection process was used at both stages to pull out each classifier’s best, most accurate features. Finally, the classifier with the highest accuracy was selected. From another perspective, statistical analysis by MINITAB software was performed to compare the time-domain features of healthy and CVD subjects. The difference between the two samples was interpreted by applying a *t*-test, which tests the null hypothesis of whether the means of the two samples are equal.

In the first step of the classification process, each time-domain characteristic’s mean and standard deviation was utilized. In the second stage, when the search space became more extensive, the search strategy could not choose the best subset of features.

Due to the fact that AF, DVT, and HF occurred less frequently than the other two cardiac conditions, there was an imbalance in class frequency. In such cases, classifiers disregarded low-frequency classes as mistakes and discarded them. Classes with low frequency were replicated, so the classifier would pay greater attention to them. As indicated in the literature, each participant was assigned a unique PPG signal and treated as a new patient.

A *p*-value between 0 and 1 for the *t*-test on the mean of time-domain features for healthy and CVD patients demonstrates the result’s significance [8].

The significance level was set at 0.05; a *p*-value > 0.05 indicates no difference in the means of the two samples, and a *p*-value below 0.05 indicates a significant difference between the means of the two samples.

Accordingly, this study induced two stages of classification. The first one classifies data into two distinct categories: healthy and CVD. In the first step of classifying the subjects, each time-domain feature’s mean and standard deviation were used. The second stage classifies CVD cases into five classes, where only the feature mean value was used, since the search method could not select the best subset of features, due to the increased search space.

In this study, seven types of classifiers were considered, i.e., decision trees (J48 and random forest), rule-based (J-Rip and PART), artificial neural network/ANN (multilayer perceptron), K-nearest neighbor (KNN), and Bayesian (Naïve Bayes) classifiers, in both stages. The attribute selection tool in WEKA selects a set of features for each classifier. This set of features is considered the best subset that improves the classifier’s performance.

## 3. Results

Each classifier’s subset of features was used to sort features into groups, and the most accurate classifier was chosen. Additionally, 10-fold cross-validation was used in this step. Table 2 and Table 3 show each classifier’s set of features and their accuracies in the first stage. Table 4 and Table 5 depict each classifier’s features and their accuracies in the second stage of the classification process. In both stages: naïve Bayes gave the highest accuracy in both stages (94.44% and 89.37%, respectively). Naïve Bayes also provided accuracies of 66.12% and 62.28%, based on PPG features only.

Sometimes it is not favorable to depend only on accuracy to evaluate the classifier. The recall is another evaluator of the classifier. The better the classifier, the higher the recall. The recall values of naïve Bayes in both stages were 0.882 in the first one and 0.806 in the second one.

## 4. Discussion

Regarding the best subset of features selected by each classifier, age dominates each classifier’s selection. This result is the same as other studies, which said that age was an essential factor in how stiff the arteries were [10]. Likewise, some CVD risk factors, such as smoking, respiratory problems, hypertension, and diabetes, appeared repeatedly in the selected feature set.

Variability analysis of PPG time-domain features assesses their degree of variation over time. These features’ variance and their means helped differentiate healthy PPG signals from CVD signals. Naïve Bayes selected the mean systolic amplitude. The fact that arterial stiffness in CVD cases changes the amplitudes of the PPG signal matches this selection. Additionally, the classifier selected the variance of diastolic time within the signal. Based on the 61% accuracy, these two features can detect CVD occurrence.

Since the variance of PPG features over time among CVD cases did not improve the classifier’s performance, only the mean of each feature represented the PPG signal. The inability to deal with the variance may be related to the fact that the search method could not select the best subset of features due to the increased search space. Arterial stiffness changes both the amplitudes and time intervals of the PPG signal, and each type of CVD affects the PPG amplitudes and intervals. So, in the second classification stage, naïve Bayes selected the CT, Notch, diastolic time, and SI, along with the APG parameters and ratios (a, b, b/a, e, e/a, (b–e)/a).

In this study, the differences between CVDs affect the CT, notch amplitude, and diastolic time selection. Finally, the SI and APG parameters are correlated with arterial stiffness, so selecting these features may relate to CVDs that include a blockage of vessels, such as CVA, DVT, and ACS.

From another point of view, naïve Bayes provides accuracies of 66.12% and 62.28%, based on the PPG features alone. The results indicate a significant effect of PPG features in classification. Both accuracy and recall in the two stages indicate an excellent evaluation of the constructed naïve Bayes classifier. The quality of the classification models can be measured by several performance measures with frequently contradictory outcomes. In [46], they compared multiple levels using various performance measures and machine learning categorization techniques. In each instance, well-established and defined techniques were utilized for the machine-learning tasks. Three datasets (acute and aquatic toxicities) were compared, and the robust yet sensitive sum of ranking differences (SRD) and analysis of variance (ANOVA) were used to evaluate the data. The effects of dataset composition (balanced versus unbalanced) and two-class versus multiclass classification scenarios were also investigated. Most performance indicators are sensitive to dataset composition, particularly in two-class classification issues. The ideal machine-learning algorithm is also very dependent on the dataset’s makeup [4]. In [50,51,52,53], the outcomes of regression performance metrics are discussed in greater detail. These studies show how important it is to validate machine learning models, which tend to work better than “traditional” regression techniques. Based on the difference between the training set and the test set’s coefficients of determination, the most reliable models are principal components and partial least squares regression.

## 5. Conclusions

As datasets, the mean and variance of the 24-time-domain feature extracted from PPG signals, demographic factors, and some information about health status were used to identify the difference between healthy human subjects and CVD patients. To classify the five types of CVDs, on the other hand, only the mean of the characteristics derived from the PPG signal and other covariates was used. In all instances, seven distinct classifiers [15,26,27,28,29,30,31,32,33] were utilized to apply feature selection and classification techniques. In the two classification stages, the naïve Bayes classifier scored the highest accuracies at 94.44% and 89.37%, respectively.

This study shows that the PPG signal can be used to find CVD by using a minimum number of PPG time-domain features, demographic factors, and some health status information. Based on these results, future work will use machine learning and deep learning algorithms to create other classification methods.

## Figures and Tables

**Figure 1 bioengineering-10-00249-f001:**
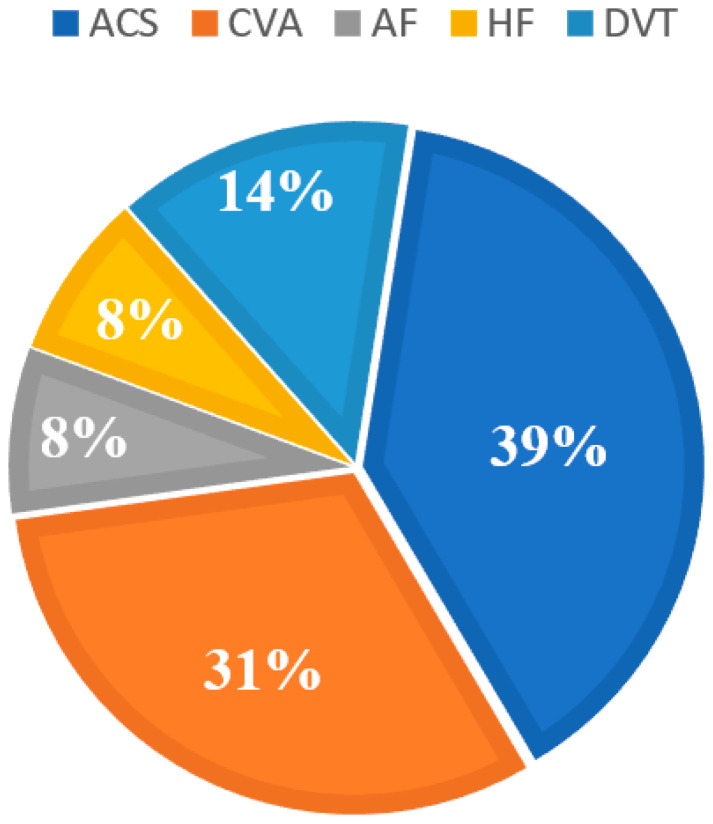
A chart of the five CVDs cases.

**Figure 2 bioengineering-10-00249-f002:**
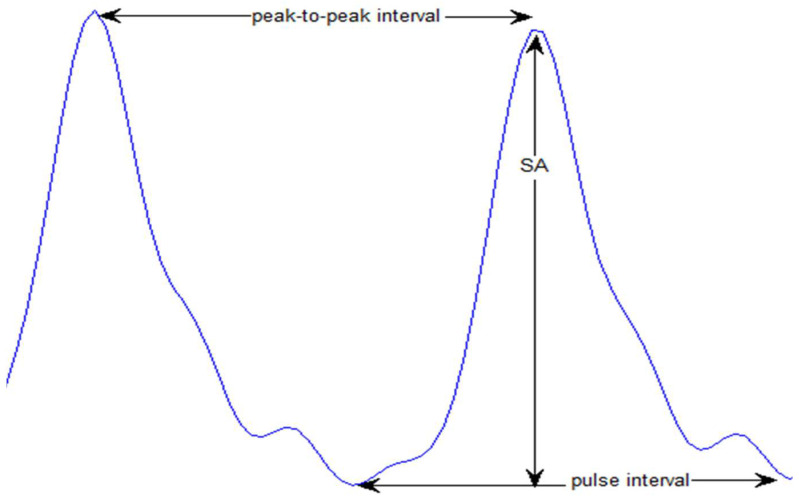
A display of SA along with peak-to- peak and pulse intervals.

**Figure 3 bioengineering-10-00249-f003:**
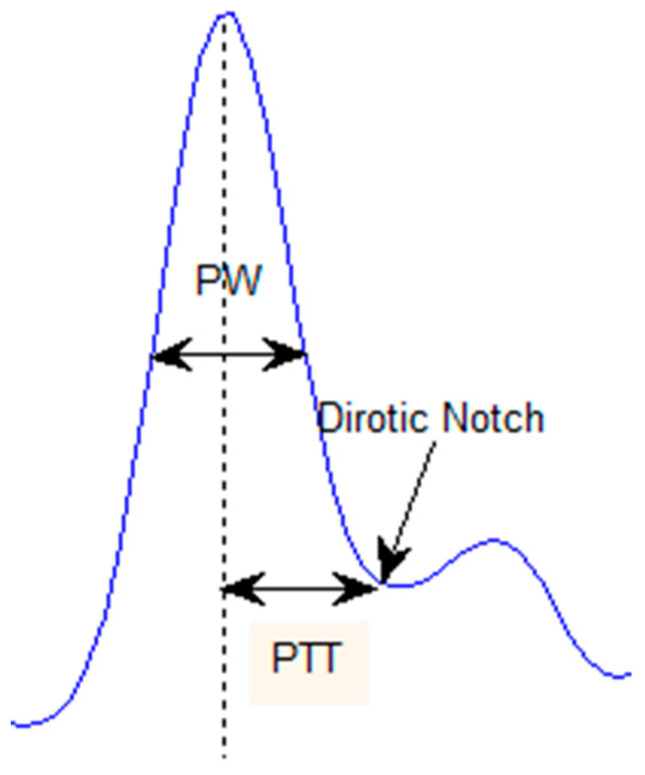
A display of PW, PTT and dicrotic notch.

**Figure 4 bioengineering-10-00249-f004:**
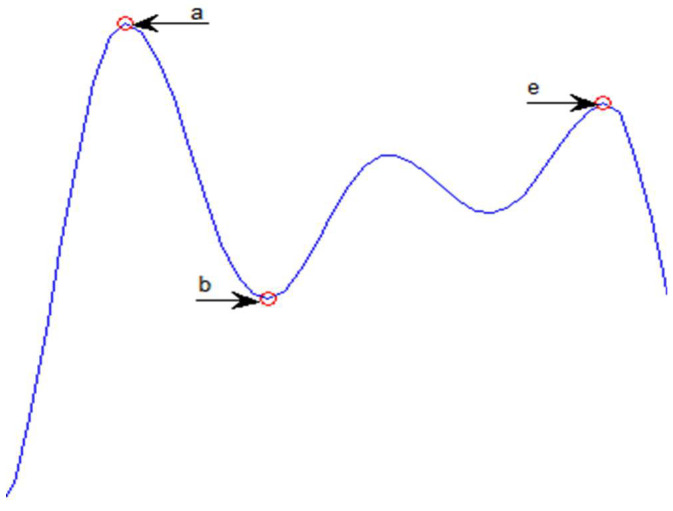
Parameters of the APG signals are: the “a” wave is the early systolic positive wave or the first positive peak of APG, the “b” wave is the early systolic negative wave or the first negative peak of APG, and the “e” wave characterizes a dicrotic notch.

**Figure 5 bioengineering-10-00249-f005:**
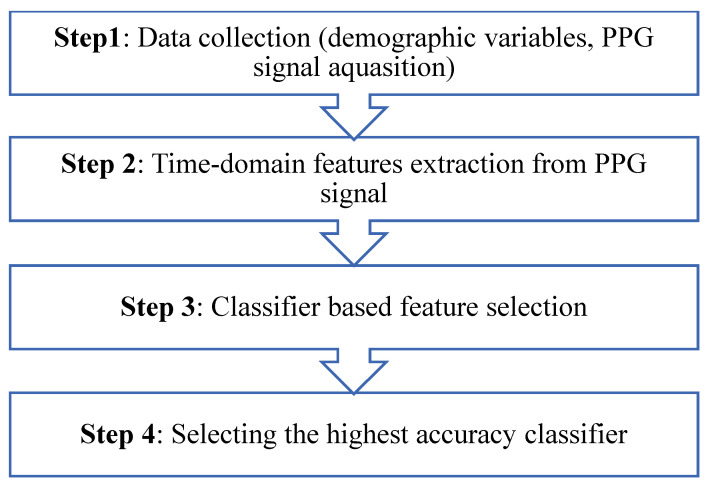
Schematic diagram of the proposed classification algorithm.

**Table 1 bioengineering-10-00249-t001:** Augmentation Indices and Time Ratios.

AI/Time Ratio	Mathematical Formula
AI of Amplitude	(SA-Dicrotic Notch Amplitude)/SA
Reflection Index (RI)	Dicrotic Notch Amplitude/SA
AI of Time	PTT/Pulse Interval
Stiffness Index (SI)	Subject’s Height/PTT
Relative Crest Time (T1 Ratio)	CT/∆T
T2 Ratio	Systolic Time/∆T
T3 Ratio	PTT/∆T

**Table 2 bioengineering-10-00249-t002:** Selected features by each classifier in the first stage.

Classifier	Selected Demographic Features	Selected PPG Features
J48	Age, weight	Pulse rate, PW, diastolic time, a, variance of e
Random forest		Pulse rate, variance of ∆T
J-Rip	Age	Pulse rate, variance of pulse interval, notch, variance of e
PART	Age, weight	Pulse rate, variance of total area, diastolic time, a, variance of e
Naïve Bayes	Age, smoking, respiratory problem, others	SA, variance of diastolic time
ANN	Age, height, BMI, gender, smoking, respiratory problem, blood pressure, others	SA, variance of SA, Notch, variance of diastolic time, AI of amplitude
KNN	Age	Variance of total area

**Table 3 bioengineering-10-00249-t003:** The accuracy of seven classifiers in the first classification stage.

Classifier	Accuracy (%)
J48	89.10
Random forest	62.94
J-Rip	85.15
PART	85.33
Naïve Bayes	94.44
ANN	89.56
KNN	82.08

**Table 4 bioengineering-10-00249-t004:** Selected features by each classifier in the second stage.

Classifier	Selected Demographic Features	Selected PPG Features
J48	Age, weight, BMI	Pulse rate, SpO_2_, SA, diastolic time, T1, b, e, e/a
Random forest		∆T
J-Rip	Age, weight, height, BMI, smoking	SpO_2_, Notch, SI, b/a
PART	Age, weight, BMI, smoking	Total area, A1, T2, AI of amplitude, b, e, e/a
Naïve Bayes	Age, height, gender, smoking, respiratory problem, blood pressure	SpO_2_, ∆T, Notch, total area, b/a, e, e/a, SI
ANN	Age, height, gender, physical activity, smoking, respiratory problem, blood pressure	CT, Notch, diastolic time, SI, a, b, b/a, e, e/a, (b–e)/a
KNN	Age, height	Notch

**Table 5 bioengineering-10-00249-t005:** The accuracy of seven classifiers in the second classification stage.

Classifier	Accuracy (%)
J48	59.36
Random forest	34.12
J-Rip	69.07
PART	72.07
Naïve Bayes	89.37
ANN	88.52
KNN	83.64

## Data Availability

Data is unavailable due to privacy of the hospital requirements and ethical restrictions.

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
