# Peer review of "Identification of Coronary Artery Diseases Using Photoplethysmography Signals and Practical Feature Selection Process"

_bioengineering, 2023, doi:10.3390/bioengineering10020249_

Round 1
Author Response
We would like to thank the reviewer for her/his in-depth analysis and careful suggestions that improved the quality of the final version of the paper and enhanced its contribution.
Photoplethysmography (PPG) is a non-invasive optical technique that can be used to detect abnormalities in the cardiovascular system by collecting relevant physiopathological parameters and has a wide range of applications in disease surveillance. In this paper, authors aimed to use data mining techniques to combine the most critical features of the PPG signal with demographic and health status variables to differentiate between healthy individuals and those with cardiovascular disease. Although this work is promising in its application, a major revision is required before its publication in Bioengineering. The details are listed as follows:
- In general, the experimental design is incomplete. The article constructs a classification model and does not perform the classification of unknown data, and lacks the validation of the model in individuals.
Authors would like to thank the reviewer for his valuable comment, the following is initially included to ensure that unknown data were classified clinically at the internal medicine unit of the hospital as follows:
’” The primary goal of this study is to distinguish between normal and abnormal cardiovascular cases by combining time-domain features of the PPG signal with a set of subject demographic characteristics. Data from 360 people (200 healthy and 160 with CVD) was collected for solid results. As for the PPG, data collection from CVD patients was carried out at the internal medicine clinic of Princess Basma Hospital. The study focused on five common types of CVDs: Acute Coronary Syndrome (ACS) (63 patients), Cerebrovascular Accident (CVA) (50 patients), Deep Vein Thrombosis (DVT) (23 patients), Heart Failure (HF), and Atrial Fibrillation (AF) (13 patients each). A chart of the five CVD cases that were studied, each with its percentage from the CVD sample, is displayed in Figure 1. ”
- What algorithm or method are used in extracting time-domain features from the PPG signal?
Response: The section that explains the extraction process is 2.2.3. PPG Signal. All subsections are numbered and the references that describe the feature extraction are included. The feature extraction process is then outlined briefly to avoid redundancy.
- Table 2 and Table 3 shows the feature set for each classifier, and the authors should give sufficient explanation and data support for the selected features of each classifier, such as the correlation coefficients of feature parameters and accuracy, ranking the importance of feature parameters.
Response: The researchers would like to thank the reviewer for his insight on this point and confirm the following:
"Selecting features or attributes is searching for and selecting the best set of features that provides the highest classification accuracy. WEKA software allows the selection of features from the dataset by applying two-step procedures: an attribute evaluator and a search method. In this study, the Classifier Subset Evaluator (ClassifierSubsetEval) method was used as the attribute evaluator. This method uses a classifier to estimate the best set of attributes. The Classifier Subset Evaluator figures out how valuable a subset of attributes is by looking at how well each feature predicts on its own and how much it overlaps with other features. Subsets of attributes that are highly correlated with the class while having low inter-correlation are preferred. At the same time, the searching method was GreedyStepWise, which performed a forward or backward search through the space of attribute subsets. For each classifier, all the features were analyzed, and the set of features that maximized the accuracy of the classifier was selected."
- Suggest adding a flow chart for the reader to better understand the feature selection process.
“ a flow chart to describe the classification process including the feature selection is included’.
“The proposed algorithm has been conducted in four steps, as shown in Figure 1. In the first step, data for healthy and CVD patients were collected. Information about the person being studied's age, gender, health, and PPG signal were among the things that were collected. MATLAB extracted time domain features from each PPG signal. In the third step, a feature selection-based classifier is used. This is done using the Waikato Environment for Knowledge Analysis (WEKA) software. This step involves two stages: the first distinguishes between healthy and CVD subjects, and the second classifies five CVD cases. A feature selection process was used at both stages to pull out each classifier's best, most accurate features. Finally, the classifier with the highest accuracy was selected. From another perspective, statistical analysis by MINITAB software was done to compare the time-domain features of healthy and CVD subjects. The difference between the two samples was interpreted by applying a t-test, which tests the null hypothesis of whether the means of the two samples are equal.”
- The figure 2 and figure 3 in your paper are a bit blurry. Please consider replacing them with clearer one.
Response: Figures are updated as requested.
- Literature 36-42 is cited in incorrect format.
Response: References are corrected to become in the right format.
Reviewer 2 Report
The paper is interesting as it provided a method to classify various types of the artery disease. The paper requires minor revisions to address the following issues.
1) The country of the vendor of the PO-80 pulse oximeter should be specified.
2) The PPG signal nature must be fully explained. PO-80 uses two wavelengths of light: 660 nm and 905 nm. Is the PPG signal the absorbance? At which wavelength?
3) The language should be improved. For example “healthy people” should be “healthy human subjects”, and “people with CVD” should be “CVD patients”.
Author Response
We would like to thank the reviewer for her/his in-depth analysis and careful suggestions that improved the quality of the final version of the paper and enhanced its contribution.
The paper is interesting as it provided a method to classify various types of the artery disease. The paper requires minor revisions to address the following issues.
- The country of the vendor of the PO-80 pulse oximeter should be specified.
Response: The data related to the PO-80 is updated with a reference for the device.
“This study employed the PO-80 pulse oximeter from Beurer Healthcare in Germany to take non-invasive PPGs (pulse rate), SpO2, and heart rate measurements.”
- The PPG signal nature must be fully explained. PO-80 uses two wavelengths of light: 660 nm and 905 nm. Is the PPG signal the absorbance? At which wavelength?
“PO-80 is a portable, rechargeable, small (L 57 mm x W 32 mm x H 30 mm), lightweight (about 42 g), and compact device that comes with "SpO2-Viewer/Manager" software that works with Windows. Using a USB connection, the software can send measured data from the device to the PC, which can be viewed in real-time. The sensor has an accuracy of +/- 2% for SpO2 in the range of 30%-100% measurement and +/- 2 beats/minute for pulse rate monitoring in the range of 30-250 beats/minute, according to its specifications.
The SpO2 sensor comprises an emitter and a detector housed in a finger-clip probe. The emitter comprises a group of light-emitting diodes (LEDs) that give off red light at 660 nm and infrared light at 905 nm. The detector, in contrast, is a silicon-based photodiode. The method used to measure PPG is conducted by inserting one finger into the finger opening of the pulse oximeter. By pressing the function button on the Po-80 sensor, the oxygen saturation of the blood's hemoglobin and the heart rate can be measured non-invasively.”
3) The language should be improved. For example “healthy people” should be “healthy human subjects”, and “people with CVD” should be “CVD patients”.
Response: “ The English language of the article is revised and corrected as requested.”
Reviewer 3 Report
Journal: Bioengineering
Title: Identification of Coronary Artery Diseases Using Photoplethysmography Signals & Practical Feature Selection Process
Authors: Amjed S. Al Fahoum*, Ansam Omar Abu Al-Haija1 and Hussam A. Alshraideh.
The idea is sound: a two-stage classification can be considered as a (small) step in the right direction.
The main finding is suspicious: “Naïve Bayes classifier got the best test accuracy of 94.44%”. There can be several reasons: the other classifiers were not optimized properly, the random fluctuations show an atypical pattern, and the design was not balanced.
Naturally more performance parameters should have been used; The authors should consult the paper:
Anita Rácz, Dávid Bajusz and Károly Héberger: Multi-Level Comparison of Machine Learning Classifiers and Their Performance Metrics. Molecules 2019; 24:2811; doi:10.3390/molecules24152811
how to consider binary and multiple classifications, and which performance parameters could be used.
It is not clear why just “the mean and variance of the 24-time domain feature” were extracted. Whole curves have more information content.
Minor errors
“the dataset [36]-[44].” i) references should be avoided in abstract, summary an conclusions; ii) it is not clear whether the references concerns for classifiers (7) or data sets.
“Higher Order Statistics (HOS).” – What is it? What is the connection of ref. [10] with this work?
“Classifier Subset Evaluator” – the term does not involve feature selection rather smaller number of samples/patients. At least some explanation is needed how does it work.
Figures 5-6 are a bar plots showing no convincing patterns. Bar plots are to be avoided according to the recommendations of the American Chemical Society: “In general, bar graphs are a waste of space and are discouraged.” [Anal. Chem. 2007, 79, 387–391].
January 18 / 2023 referee:
Author Response
We would like to thank the reviewer for her/his comments which improved the quality of the paper and made a substantial contribution to its final form.
The idea is sound: a two-stage classification can be considered as a (small) step in the right direction.
The main finding is suspicious: “Naïve Bayes classifier got the best test accuracy of 94.44%”. There can be several reasons: the other classifiers were not optimized properly, the random fluctuations show an atypical pattern, and the design was not balanced.
Naturally more performance parameters should have been used; The authors should consult the paper:
Anita Rácz, Dávid Bajusz and Károly Héberger: Multi-Level Comparison of Machine Learning Classifiers and Their Performance Metrics. Molecules 2019; 24:2811; doi:10.3390/molecules24152811
how to consider binary and multiple classifications, and which performance parameters could be used.
Response: Authors would like to thank the reviewer for pointing out to this point for further discussion, at different parts of the manuscript these points were considered:
“Classification algorithms based on machine learning are commonly employed to anticipate and categorize the many features of biomedical signals and molecules, such as cardiac anomalies, toxicity or biological activity [46-50]. Three data sets were generated to test different classification models based on machine learning [46]. Further, they applied many classification performance indicators from many fields. To compare the two groups, they used a novel method that was based on traditional chemometric techniques like the sum of ranking differences (SRD) and analysis of variance (ANOVA) [50].”
“The quality of classification models can be measured by several performance measures with frequently contradictory outcomes. In [46], they compared multiple levels using various performance measures and machine learning categorization techniques. In each instance, well-established and defined techniques were utilized for the machine-learning tasks. Three datasets (acute and aquatic toxicities) were compared, and the robust yet sensitive sum of ranking differences (SRD) and analysis of variance (ANOVA) were used to evaluate the data. The effects of dataset composition (balanced versus unbalanced) and 2-class versus multiclass classification scenarios were also investigated. Most performance indicators are sensitive to dataset composition, particularly in 2-class classification issues. The ideal machine-learning algorithm is also very dependent on the dataset's makeup [46]. In [52-54], the outcomes of regression performance metrics are discussed in greater detail. These studies show how important it is to validate machine learning models, which tend to work better than "traditional" regression techniques. Based on the difference between the training set and the test set's coefficients of determination, the most reliable models are principal components and partial least squares regression.”
It is not clear why just “the mean and variance of the 24-time domain feature” were extracted. Whole curves have more information content.
Response: It sounds like this point was not enough clear, the following paragraph was added:
“In the first step of the classification process, each time domain characteristic's mean and standard deviation are utilized. In the second stage, when the search space got more extensive, the search strategy could not choose the best subset of features.
Due to the fact that AF, DVT, and HF occurred less frequently than the other two cardiac conditions, there was an imbalance in class frequency. In such cases, classifiers disregard low-frequency classes as mistakes and discard them. Classes with low frequency were replicated so the classifier would pay greater attention to them. As indicated in the literature, each participant was assigned a unique PPG signal and treated as a new patient.
A p-value between 0 and 1 for the t-test on the mean of time-domain features for healthy and CVD patients demonstrates the result's significance [50].
The significance level is set at 0.05; a p-value > 0.05 indicates no difference in the means of the two samples, and a p-value below 0.05 indicates a significant difference between the means of the two samples.”
Minor errors
“the dataset [36]-[44].” i) references should be avoided in abstract, summary an conclusions; ii) it is not clear whether the references concerns for classifiers (7) or data sets.
Response: References are removed from the abstract.
“Higher Order Statistics (HOS).” – What is it? What is the connection of ref. [10] with this work?
HOS are those statistics with values higher than 2:
To make it clearer:
“Yousefi et al. [10] found a way to find PVC automatically by combining the chaotic nature of the PPG signal with Higher Order Statistics (HOS). They used several chaotic and statistical features, such as the Lyapunov exponent, skewness, kurtosis, fuzzy entropy, and spectral entropy, which were taken from the signals. Principal component analysis (PCA) was used to figure out which data points should be used to group the data.“
“Classifier Subset Evaluator” – the term does not involve feature selection rather smaller number of samples/patients. At least some explanation is needed how does it work.
Response: For better clarification, the following paragraph is added.
“The Classifier Subset Evaluator figures out how valuable a subset of attributes is by looking at how well each feature predicts on its own and how much it overlaps with other features. Subsets of attributes that are highly correlated with the class while having low inter-correlation are preferred.”
Figures 5-6 are a bar plots showing no convincing patterns. Bar plots are to be avoided according to the recommendations of the American Chemical Society: “In general, bar graphs are a waste of space and are discouraged.” [Anal. Chem. 2007, 79, 387–391].
Response: In their place, bar plots were replaced with tables.
Round 2
Reviewer 1 Report
Papers need to be slightly modified before they are accepted by journals:References are coded consecutively in Arabic numerals according to the order in which they appear in the text.
Reviewer 3 Report
I read the entire manuscript and the yellow markings. I am satisfied with the completions. Now the MS corresponds to highest standards. I suggest acceptance as is.